# Predictors of Depression in Elderly According to Gender during COVID-19: Using the Data of 2020 Community Health Survey

**DOI:** 10.3390/healthcare12050551

**Published:** 2024-02-27

**Authors:** Hye-Jung Jun, Kyoung-Mi Kim

**Affiliations:** 1Department of Nursing, Busan Women’s College, Busan 47228, Republic of Korea; junhj70@hanmail.net; 2College of Nursing, Kosin University, Busan 49267, Republic of Korea

**Keywords:** COVID-19, aged, depression, gender

## Abstract

Background: This study aimed to examine factors influencing depression according to gender in people during COVID-19. Methods: This study was conducted on 61,147 elderly individuals over the age of 65 who participated in the 2020 Community Health Survey (CHS). Data analysis was conducted using SAS 9.4. Results: Elderly females had a higher perceived fear of COVID-19 than males. The common factors affecting depression in elderly individuals were age, monthly income, economic activity, stress, subjective health status, and social support. Among elderly women, changes in residential areas and daily life due to COVID-19 were identified as factors affecting depression. Conclusions: Therefore, during COVID-19, it was necessary to provide customized depression relief programs for the elderly, and it was necessary to find ways for them to positively perceive their health status and increase healthcare efficacy. In the future, it is necessary to pay attention to elderly women living in rural areas and make efforts to ensure that their daily lives are not interrupted by infectious diseases.

## 1. Introduction

Since the World Health Organization declared COVID-19 a global pandemic in March 2020, over 60% of Koreans have been impacted. Globally, the elderly face a higher risk of contracting the virus. In the U.S., about 80% of COVID-19-related deaths have occurred in those over 65 [1]. In Australia, the majority of fatalities have been among people older than 70 [2]. The mortality rate for older individuals is also notably higher in Asia [3]. In Korea, the mortality rates for those in their 60s, 70s, and 80s have been 0.12%, 0.45%, and 1.94%, respectively, with the elderly being particularly vulnerable [4,5,6,7]. Thus, it is crucial for society to focus on the elderly, who are most at risk from such infectious diseases.

As individuals age, their health typically deteriorates, a concern particularly relevant to the elderly, who often face limitations in activities due to chronic diseases. This, coupled with a shrinking social circle and increased loneliness following the loss of close ones, significantly impacts their well-being [6]. The advent of COVID-19 brought about further challenges, imposing restrictions on daily life and social interactions due to social distancing measures [8,9]. These constraints have notably affected the elderly, with reports indicating a rise in depression, primarily caused by the curtailment of daily activities and social disconnection [5]. Therefore, it is crucial to monitor depression rates in this demographic, especially in light of potential future pandemics. Developing strategies to effectively address and alleviate depression among the elderly is thus a pressing need.

Depression in the elderly can also lead to suicidal thoughts and attempts [10]. In fact, the suicide rate among people aged over 65 in Korea has been two to three times higher than that among those under 65 years of age over the past 8 years. The suicide rate stands at approximately 65 per 100,000 people, a figure considered quite high even in comparison to the OECD (Organization for Economic Cooperation and Development) average [11]. Therefore, it is crucial to proactively assess depression levels among the elderly and identify high-risk groups to prevent the progression of depression into suicidal tendencies. Infectious diseases, such as COVID-19, are considered risk factors that can exacerbate the fear of illness and depression in the elderly. Given that new infectious diseases can resurface at any time, it is meaningful to assess whether the elderly population during the COVID-19 era is experiencing depression as a preventive measure against its development or exacerbation.

Depression in the elderly is known to vary depending on gender [12]. Particularly, among elderly women, the severity of depression tends to increase as their health and daily living conditions deteriorate. They are also more likely to experience suicidal thoughts and even attempt suicide due to depression. In contrast, among elderly men, depression often takes on a chronic nature, leading to a heightened risk of depressive symptoms and suicidal thoughts, but fewer suicide attempts [13]. Furthermore, in the era of COVID-19, elderly women living alone have seen a decrease in their levels of social activity, while their interactions with family members have increased significantly more than in elderly men [14]. Therefore, the objective of this study is to provide fundamental data for tailored interventions for each gender, differentiating between elderly males and females in the context of the COVID-19 era and assessing the pandemic’s impact on depression.

Previous studies have indicated that elderly individuals experienced higher levels of depression during the COVID-19 era due to restrictions that limited their social circles and participation in social activities [5]. It was observed that depression was less prevalent among elderly individuals who had many close relationships and frequently communicated with their children [4]. Essentially, most prior studies assessed depression during the COVID-19 era within the context of social relationships. However, there has been no study that distinguishes between genders and analyzes how subjective health status, stress, changes in daily life, and the fear of COVID-19 infection contribute to depression.

This study focuses on individuals aged 65 years or older, both men and women, during the COVID-19 era. It aims to identify sociodemographic characteristics, assess subjective health status, evaluate stress levels, examine social support, analyze changes in daily life, and assess the impact of the fear of COVID-19 infection on depression. Through this analysis, we aim to provide essential data for the development of programs aimed at mitigating depression during future global pandemics.

## 2. Materials and Methods

### 2.1. Study Design and Data Source

This study used raw data from the Korea Disease Control and Prevention Agency’s 2020 Community Health Survey (CHS). The Community Health Survey is a survey that has been conducted every year at public health centers across the country since 2008, with the purpose of identifying local health statistics for establishing and evaluating local healthcare plans. Moreover, it is a survey that assesses not only health behaviors, morbidity, unmet medical care, and social and physical environments but also individual and physical environments through health-related questions at the regional level. The target population lives nationwide in cities, counties, and districts from July 2020, while the survey targets adults aged 19 or older living in the selected sample households at the time of the survey. Sampling represents a complex design, whereby sampling points are first extracted based on the number of households and according to housing type in Tong/Ban/Ri. The second step comprises selecting the final sample households using a systematic sampling method. The community health survey was conducted from 16 August to 31 October 2020 by trained investigators directly visiting sample households and conducting 1:1 interviews with subjects using a laptop loaded with 142 questions in 18 areas (computer-assisted personal interviewing). Among the 229,269 people who participated in the 2020 Community Health Survey, 72,812 were 65 years old, and 11,665 people were excluded owing to missing values, meaning 61,147 people were used in the final analysis. Therefore, the Community Health Survey is effective in collecting health-related information from various groups within the community, as it employs a random sampling method that includes diversity in age, gender, and socioeconomic status.

### 2.2. Variables

The sociodemographic characteristics of the subjects included age, education level, average monthly household income, economic activity, residential area, stress, and subjective health status. Age was categorized into three groups: 65–74, 75–84, and 85 or older. Educational attainment was recategorized into “uneducated”, “elementary school graduate or higher”, and “middle school graduate or higher”. Average monthly household income was classified as follows: <KRW 1,000,000, KRW 1,000,000–2,990,000, and >KRW 3,000,000. Economic activity status was determined by current engagement in economic activity. Residential areas were divided into urban and rural areas. The subjective level of perceived stress was formulated as a single question, asking, “How much stress do you feel in your daily life?”, while the answers consisted of “I feel it very much”, “I tend to feel it a lot”, “I tend to feel it a little”, and “I hardly feel it”. In this study, it was recategorized as “do not feel”, “average”, and “feel a lot”. Subjective health status was presented as a single question, asking “How do you think your health is in general?”, while the answers consisted of “very good”, “good”, “average”, “bad”, and “very bad”. Again, in this study, the answer categories were reclassified as “bad”, “average”, and “good”.

Social support also featured as a question, which enquired over the number of relatives, friends, neighbors, and acquaintances, including brothers and sisters, the individual was close to (with whom you can confide). It was reclassified as none if there is no support, or more than one.

The daily life change tool was the question “If 100 points were the state of daily life before the COVID-19 epidemic and 0 points indicated that daily life was completely stopped, what is your current state?” A score of 0 points indicated complete stoppage, and a score of 100 meant no change.

Items regarding fear of infection due to the COVID-19 epidemic included “I am worried about being infected with COVID-19” and “I am worried that I could die if I become infected with COVID-19”. This was answered on a 5-point scale comprising “very true”, true”, “moderate”, “not very true”, and “not at all true”.

The existence of depression consisted of a yes/no answer question: “Have you ever felt so sad or hopeless that it interferes with your daily life for more than two weeks in a row over the past year?”

### 2.3. Data Collection Process

Prior consent for the community health survey was obtained from all survey participants before data collection commenced. The tools and survey procedures were carried out following approval by the Korea Disease Control and Prevention Agency’s Medical Research Ethics Review Committee. This study received and analyzed anonymized data with personal identifying information removed, in full compliance with the Korea Disease Control and Prevention Agency’s regulations governing procedures for the disclosure of raw data.

### 2.4. Statistical Analysis

Research data were analyzed using the SAS 9.4 program (SAS Institute Inc., Cary, NC, USA). Descriptive statistical analysis was performed to determine the subjects’ sociodemographic characteristics, social support, changes in daily life due to COVID-19, fear of COVID-19 infection, and degree of depression. The Rao-Scott x2 test was performed to determine differences in depression according to the characteristics of each subject. Complex sample logistic regression analysis was conducted to identify factors affecting the subjects’ depression. All analyses were conducted using sample weights from the Community Health Survey.

## 3. Results

### 3.1. Sociodemographic Characteristics, Changes in Daily Life Due to COVID-19, and Level of Fear of Infection

This study comprised 24,969 men (40.83%) and 36,178 women (59.17%), with 31,758 (51.94%) individuals identified in the largest age group of 65–74 years old. As for education level, 16,459 people (26.92%) had no education, 20,541 people (33.59%) had graduated from elementary school, and 24,147 people (39.49%) had graduated from middle school or higher. The average monthly household income was KRW 180.10 ± 180.52, while 21,824 people (35.69%) were economically active and 39,323 people (64.31%) were not economically active. Residential areas consisted of 26,735 people (43.72%) residing in the city, and 34,412 people (56.28%) living in the countryside. Regarding the subjective perception of stress, 26,224 people (42.89%) did not feel stressed, 26,270 people (42.96%) felt moderate stress, and 8653 people (14.15%) felt very stressed. Subjective health status was good for 19,599 people (32.05%), fair for 24,127 people (39,46%), and poor for 17,421 people (28.49%). A total of 15,202 people (24.86%) had no social support network, while 45,945 people (75.145) had support from at least one person. There were 3283 people (5.36%) who had experienced depression, and 57,864 people (94.63%) who had not experienced depression (Table 1).

The average change in the subject’s daily life due to COVID-19 was 59.3 ± 24.2 out of 100, and the average fear of becoming infected by COVID-19 was 3.80 ± 1.00 out of 5 (Table 2).

### 3.2. Differences in Depression Depending on the Subject’s Characteristics

Depression was determined by age (χ^2^ = 316.09, *p* < 0.001), education level (χ^2^ = 8703.51, *p* < 0.001), average monthly household income (χ^2^ = 1617.23, *p* < 0.001), and economic activity (t = 970.78, *p* < 0.001), alongside residential area (t = 50.61, *p* < 0.001), subjective stress (χ^2^ = 267.72, *p* < 0.001), subjective health status (χ^2^ = 1564.88, *p* < 0.001), and fear of COVID-19 infection (χ^2^ = −20.62, *p* < 0.001), which demonstrated statistically significant differences (Table 3).

### 3.3. Factors Affecting the Subject’s Depression

Multiple logistic regression analysis was conducted to identify factors affecting the subjects’ depression. As a result of the analysis, factors influencing depression in elderly men were identified as age, average monthly household income, economic activity, subjective stress, subjective health status, and social support (Table 4).

Compared to male seniors aged 65–74 years, the experience of depression decreased by 0.85 times (95% CI: 0.73–0.98) in male seniors aged 75–84 years. Compared to older men with an average monthly household income of <KRW 1,000,000, older men with an average monthly household income of between KRW 1,000,000 and KRW 3,000,000 are 0.74 times more likely to feel depressed (95% CI: 0.66–0.86), and those with an average monthly household income of more than KRW 3,000,000 are 0.57 times more likely to feel depressed (95% CI: 0.46–0.71). Thus, elderly men are less likely to feel decreased with higher monthly household incomes. Additionally, depression among economically inactive elderly men increased 1.99 times compared to economically active elderly men (95% CI: 1.70–2.35). Moreover, depression increased 3.05 times (95% CI: 2.51–3.70) in older men with average stress and 10.98 times (95% CI: 9.00–13.40) in older men who felt a lot of subjective stress, compared to older men who do not feel subjective stress. Depression increased 1.24 times (95% CI: 1.04–1.49) in elderly men with average health and 2.37 times (95% CI: 1.97–2.84) in elderly men with poor subjective health, compared to elderly men with good subjective health. Finally, older men with one or more supportive people showed a 0.86-fold (95% CI: 0.75–0.99) decrease in depression, compared to older men without any social support network.

Factors influencing depression in elderly women were identified as age, average monthly household income, economic activity, residential area, subjective stress, subjective health status, social support, and changes in daily life due to COVID-19 (Table 5).

Regarding age, depression decreased by 0.86 times (95% CI: 0.86–0.95) in women aged 75–84 years and by 0.75 times (95% CI: 0.63–0.90) in women aged 85 years or older, compared to women aged 65–74 years. In terms of average monthly household income, compared to elderly women earning <KRW 1,000,000, elderly women earning between KRW 1,000,000 and KRW 3,000,000 were 0.66 times more likely to feel depressed (95% CI: 0.60–0.73), and older women earning >KRW 3,000,000 were 0.56 times more likely to feel depressed (95% CI: 0.48–0.64), which was similar to in males, whereby higher monthly household incomes reduced feelings of depression. Depression among economically inactive elderly women increased 1.37 times (95% CI: 1.23–1.52) compared to economically active elderly women. Moreover, depression among elderly women living in the city decreased by 0.88 times (95% CI: 0.80–0.97) compared to elderly women living in rural areas. Depression increased by 1.21 times (95% CI: 2.09–2.71) in elderly women who felt average and 9.23 times (95% CI: 8.11–10.50) among elderly women who felt a lot of subjective stress, compared to elderly women who did not feel subjective stress. Furthermore, depression increased 1.21 times (95% CI: 1.06–1.39) for elderly women with average health status and 2.02 times (95% CI: 1.77–2.30) for elderly women with poor subjective health, compared to elderly women with good subjective health. While depression among elderly women with one or more persons of support decreased by 0.88 times (95% CI: 0.80–0.97), compared to elderly women without any social support network. Finally, depression was confirmed to be reduced by 0.99 times (95% CI: 0.99–0.99) in elderly women whose daily lives were actively changing compared to elderly women whose daily lives were completely halted due to COVID-19.

## 4. Discussion

In this study, a significant difference was observed in the level of fear of COVID-19 infection between male and female seniors. It was confirmed that women had a higher level of fear than men, with women scoring 3.89 points compared to men’s score of 3.71 points. These research findings align with a previous study [15] that identified similar patterns of COVID-19 fear among the general adult population, with women expressing higher levels of fear than men. This outcome may be attributed to the fact that in this study, 43% of elderly male respondents reported being economically active, whereas only 30.7% of elderly female respondents indicated economic activity. Considering previous research indicating that individuals with economic instability tend to experience more fear of COVID-19 [16], it can be inferred that many elderly women, due to their lack of economic activity, are more vulnerable to COVID-19 risks, potentially leading to increased concerns and fears.

Among the factors influencing depression in older men and women during COVID-19, residential areas and changes in daily life due to the pandemic were factors identified as significant only for women. In this study, the residential area did not emerge as a predictor of depression in male seniors. However, it was established that depression was less prevalent among elderly women living in urban areas compared to those residing in rural communities. Interestingly, these findings diverge from previous research [17,18]. The discrepancy can be attributed to the fact that data collection in prior studies occurred before the onset of the COVID-19 pandemic when rural elderly individuals generally engaged in more physical activity, contributing to lower depression rates. In contrast, this study was conducted during the pandemic, characterized by restricted physical activity due to social distancing and lockdown measures affecting the elderly. Notably, elderly individuals living in urban areas often had greater access to medical care than their rural counterparts, aligning with the results of this study, wherein elderly women in urban areas exhibited lower levels of depression. Considering earlier research that has associated higher levels of depression with unmet medical needs [5], it is imperative that future responses to new infectious disease outbreaks pay closer attention to assessing depression among elderly women in rural areas. Strategies should aim to enhance their access to medical care and overall satisfaction with healthcare services, ultimately mitigating depression.

Additionally, it was established that changes in daily life due to COVID-19 significantly influenced depression among elderly women. In comparison to elderly women whose daily routines were entirely disrupted by COVID-19, those who remained active in their daily lives exhibited a 0.99 times reduction in depression levels. A prior study also corroborated similar findings regarding depression among the elderly during the COVID-19 pandemic, indicating a negative correlation between engagement in daily activities and social participation with depression [5]. In this study, active adjustments to daily life emerged as a key factor impacting depression among elderly women, particularly when contrasted with elderly men. This underscores the necessity for future initiatives aimed at reinvigorating the social engagement of elderly women through various means, such as online platforms, especially in the context of global pandemics.

In this study, concerning the elderly male community during COVID-19, depression was observed to decrease in the middle-aged elderly group (75–84 years old) in comparison to the younger elderly group (65–74 years old). Similarly, among elderly women, depression decreased in both the middle-aged (75–84 years old) and the higher-aged (85 or older) groups when compared to the early-aged (65–74 years old) group. These findings contradict previous studies that suggested older individuals experience higher levels of depression [19]. In essence, this research underscores that elderly individuals aged 64 to 74 are particularly susceptible to depression. A prior study also reported severe depression as one of the predictors of falls among elderly people aged 65 to 74 [20]. Hence, there is a pressing need to establish a system for assessing and managing depression among individuals in the 64 to 74 age group.

In this study, it was established that the income level of elderly men and women significantly influenced depression. Specifically, individuals with a monthly household income between KRW 1 million and KRW 3 million, as well as those with an income exceeding KRW 3 million, experienced lower levels of depression compared to the group earning less than KRW 1 million. Furthermore, it was confirmed that depression rates increased among economically inactive males and females. These findings align with the results of a previous study [5], which identified average household monthly income as a predictive factor for depression in the elderly. Moreover, it is worth noting that the elderly population in Korea is more likely to experience economic vulnerability due to lower income levels compared to both younger and older generations. Low income can lead to economic stress and reduced healthcare spending, thereby contributing to higher levels of depression among the elderly [21,22,23]. Therefore, it is imperative for the government to address and mitigate poverty among the elderly population, actively facilitating employment opportunities to improve their income.

In this study, it was observed that compared to seniors who do not experience subjective stress, the average senior experienced more depression, while those who felt extensively stressed exhibited even higher levels of depression. Furthermore, depression was found to be more prevalent among senior citizens with poor subjective health compared to those with good subjective health. These findings align with previous research [23], which has highlighted a strong correlation between the subjective health status of older individuals and depression. Given that this study has confirmed the significant influence of subjectively perceived stress and health levels on depression among the elderly, it underscores the importance of community nurses maintaining a positive perception of elderly individuals’ health status when providing care in the future. This approach should be guided by a belief in the potential to address and alleviate these issues. Therefore, efforts to raise awareness are crucial, particularly during periods when certain infectious diseases, such as COVID-19, are prevalent. Additionally, there is a heightened need for comprehensive initiatives aimed at enhancing health awareness among the elderly through various means.

In this study, it was confirmed that depression decreased in older people with one or more people offering support compared to older people with no social support network. This study can be seen in a similar context to a study [24,25,26] on elderly people during COVID-19, which found that emotional support from others acted as a buffering effect against COVID-19 leading to the onset of depression. In addition, a study on elderly people in the COVID-19 era presented the same results as those, which showed that elderly people receiving family support reduced depression [6]. Considering that during COVID-19, depression in seniors living alone decreased when they met with close people more often and had children or family members who contacted them more than once a week by phone [4], it is a belief that nurses should pay close attention to check the support system and either strengthen it or encourage the formation of one. In this study, social isolation caused by reasons such as COVID-19 has been identified as a contributing factor to depression among the elderly. Therefore, based on this finding, there is a need for various policies aimed at preventing social isolation among the elderly, and efforts to prevent depression in this population require active intervention at the national level. According to prior research [27], there is no one-size-fits-all approach to addressing loneliness or social isolation; interventions need to be tailored to suit the needs of individuals, specific groups, or the degree of loneliness experienced. Thus, to address social isolation among the elderly, personalized interventions based on the needs and conditions of the elderly are required.

This study is meaningful because it identified factors affecting depression in older men and women in the COVID-19 era. However, this study is a secondary data analysis study based on existing community health survey data, and some limitations exist in the interpretation of its results. Indeed, this study is a cross-sectional study and cannot clearly determine the causal relationship between subjective health status, stress, social support, changes in daily life, fear of COVID-19, and depression. Additionally, a limitation of this study is that it did not investigate beyond the frequency of social support to assess the context of interactions, such as whether they occurred face-to-face, non-face-to-face, or online. To assess confidence in healthcare access, it can be considered to measure various aspects such as satisfaction with medical services, ease of access, and perception of the quality of healthcare services. In future research, it is necessary to include a variety of question types, such as Likert scales and open-ended questions, to deeply understand the experiences and perceptions of the participants. In this study, depression was assessed dichotomously; however, future research should consider using a variety of question formats to more thoroughly assess depression in the elderly. Although this tool is not designed to assess the presence or severity of depression, which could be a limitation of this study, it is significant in that it checks whether elderly individuals have recently experienced feelings of depression and uses this as a basis to predict depression among the elderly during a pandemic. While this may not be a novel discovery, the significance of this study lies in moving beyond previous research that identified depression in the elderly, by distinguishing depression in the elderly according to gender. Despite these limitations, this study analyzed data representative of elderly men and women at the national level and identified factors influencing their depression, thereby providing basic data for the development of future policies and action plans related to the mental health of the elderly.

## 5. Conclusions

This study uses data from the 2020 Community Health Survey to identify factors affecting depression in older men and women during the COVID-19 pandemic and to provide evidence for future customized depression relief nursing interventions in older men and women.

As a result of this study, it was confirmed that older men and women perceived their fear of COVID-19 to be higher. The common factors influencing depression in older men and women were age, monthly income, economic activity, stress, subjective health status, and social support networks. In particular, among elderly women, changes in residential areas and daily life due to COVID-19 were identified as factors influencing depression.

Therefore, in future global pandemics, it is necessary to provide customized depression relief programs for the elderly in the early, middle, and late stages, and it is necessary to find ways to positively perceive their health status and increase health management efficacy. During the period of infectious diseases, efforts are needed to alleviate depression, focusing on elderly women living in rural areas, where unmet medical care may occur. In addition, daily life is not interrupted due to infectious diseases, and daily life can be improved through other methods, such as non-face-to-face contact. Efforts are required to minimize change as much as possible. To support the elderly, it is essential to expand online platform-based counseling and education programs and provide continuous support using telephone or messaging services. Additionally, developing programs that offer opportunities for active participation to prevent social isolation and strengthen social support networks among seniors is crucial. These programs should aim to enhance social networking and community engagement among the elderly. Ultimately, it is believed that this will help maintain a healthy life for the elderly by reducing depression in both male and female seniors.

## Figures and Tables

**Table 1 healthcare-12-00551-t001:** Characteristics of the subjects (N = 61,147).

Characteristics	Categories	n	(%)	M ± SD
Gender	Male	24,969	(40.83)	
	Female	36,178	(59.17)	
Age (year)	65–74	31,758	(51.94)	74.32 ± 6.50
	75–84	24,141	(39.48)	
	≥85	5248	(8.58)	
Education level	None	16,459	(26.92)	
	Elementary school	20,541	(33.59)	
	≥Middle school	24,147	(39.49)	
Monthly income (KRW 10,000)	<100	24,153	(39.50)	180.10 ± 180.52
	100–300	25,769	(42.14)	
	≥300	11,225	(18.36)	
Economic activity	Yes	21,824	(35.69)	
	No	39,323	(64.31)	
Residence	City	26,735	(43.72)	
	Rural	34,412	(56.28)	
Stress	None	26,224	(42.89)	
	Normal	26,270	(42.96)	
	Very stressed	8653	(14.15)	
Subjective health status	Good	19,599	(32.05)	
	Normal	24,127	(39.46)	
	Bad	17,421	(28.49)	
Social support	None	15,202	(24.86)	
	≥1	45,945	(75.14)	
Depression	Yes	3.283	(5.36)	
	No	57,864	(94.63)	

**Table 2 healthcare-12-00551-t002:** Level of changes in daily life due to COVID-19, and fear of COVID-19 infection (N = 61,147).

Characteristics	Mean	SD	Min	Max
Changes in daily life due to COVID-19	59.3	24.2	0	100
Fear of COVID-19 infection	3.8	1.0	1	5

**Table 3 healthcare-12-00551-t003:** Differences in general characteristics (N = 61,147).

Characteristics	Categories	Male (N = 24,969)	Female (N = 36,178)	χ^2^ or t	*p*
		N (%)	N (%)		
Age (year)	65–74	13,903 (55.7)	17,855 (49.4)	316.09	<0.001
	75–84	9377 (37.6)	14,764 (40.8)		
	≥85	1689 (6.7)	3559 (9.8)		
Education level	None	2535 (10.2)	13,924 (38.5)	8703.51	<0.001
	Elementary school	7634 (30.6)	12,907 (35.7)		
	≥Middle school	14,800 (59.2)	9347 (25.8)		
Monthly income (KRW 10,000)	<100	7478 (29.9)	16,675 (46.1)	1617.23	<0.001
	100–300	12,074 (48.4)	13,695 (37.8)		
	≥300	5417 (21.7)	5808 (16.1)		
Economic activity	Yes	10,726 (43.0)	11,098 (30.7)	970.78	<0.001
	No	14,243 (57.0)	25,080 (69.3)		
Residence	City	11,346 (45.4)	15,389 (42.5)	50.61	<0.001
	Rural	13,623 (54.6)	20,789 (57.5)		
Stress	None	11,355 (45.5)	14,869 (41.1)	267.72	<0.001
	Normal	10,730 (43.0)	15,540 (43.0)		
	Very stressful	2884 (11.5)	5769 (15.9)		
Subjective health status	Good	9965 (39.9)	9634 (26.6)	1564.88	<0.001
	Normal	9662 (38.7)	14,465 (40.0)		
	Bad	5342 (21.4)	12,079 (33.4)		
Social support	None	6212 (24.9)	8990 (24.8)	0.01	0.934
	≥1	18,757 (75.1)	27,188 (75.2)		
Changes in daily life due to COVID-19		59.41 ± 24.05	59.27 ± 24.22	0.69	0.492
Fear of COVID-19 infection		3.71 ± 1.04	3.89 ± 1.00	−20.62	<0.001

**Table 4 healthcare-12-00551-t004:** Factors influencing depression in elderly males (N = 61,147).

Characteristics	Categories	Unstandardized Coefficients	SE	Wald	OR (95% CI)	*p*
Age (ref = 65–74 years)	75–84	−0.167	0.075	4.991	0.85 (0.73–0.98)	0.026
	≥85	−0.236	0.137	2.987	0.79 (0.60–1.03)	0.084
Education level (ref = none)	Elementary school	−0.144	0.110	1.710	0.87 (0.70–1.07)	0.191
	≥Middle school	−0.138	0.108	1.636	0.87 (0.71–1.08)	0.201
Monthly income (KRW 10,000) (ref ≤ 100)	100–300	−0.304	0.076	15.858	0.74 (0.66–0.86)	<0.001
	≥300	−0.558	0.110	25.780	0.57 (0.46–0.71)	<0.001
Economic activity (ref = yes)	No	0.692	0.082	71.307	1.99 (1.70–2.35)	<0.001
Residence (ref = rural)	City	−0.072	0.072	1.017	0.93 (0.81–1.07)	0.313
Stress (ref = none)	Normal	1.115	0.099	127.451	3.05 (2.51–3.70)	<0.001
	Very stressful	2.396	0.102	557.920	10.98 (9.00–13.40)	<0.001
Subjective health status (ref = good)	Normal	0.219	0.093	5.548	1.24 (1.04–1.49)	0.019
	Bad	0.863	0.093	85.859	2.37 (1.97–2.84)	<0.001
Social support (ref = none)	≥1	−0.151	0.073	4.225	0.86 (0.75–0.99)	0.040
Changes in daily life due to COVID-19		−0.001	0.001	0.510	0.99 (0.10–1.00)	0.475
Fear of COVID-19 infection		−0.002	0.033	0.003	0.99 (0.94–1.07)	0.956

**Table 5 healthcare-12-00551-t005:** Factors influencing depression in elderly females (N = 61,147).

Characteristics	Categories	Unstandardized Coefficients	SE	Wald	OR (95% CI)	*p*
Age (ref = 65–74 years)	75–84	−0.156	0.051	9.289	0.86 (0.77–0.95)	0.002
	≥85	−0.288	0.090	10.207	0.75 (0.63–0.90)	0.001
Education level (ref = none)	Elementary school	−0.026	0.055	0.217	0.98 (0.88–1.09)	0.641
	≥Middle school	0.078	0.066	1.402	1.08 (0.95–1.23)	0.237
Monthly income (KRW 10,000) (ref ≤ 100)	100–300	−0.417	0.051	66.127	0.66 (0.60–0.73)	<0.001
	≥300	−0.589	0.074	62.642	0.56 (0.48–0.64)	<0.001
Economic activity (ref = yes)	No	0.312	0.054	33.156	1.37 (1.23–1.52)	<0.001
Residence (ref = rural)	City	−0.129	0.049	7.083	0.88 (0.80–0.97)	0.008
Stress (ref = none)	Normal	0.866	0.066	170.209	2.38 (2.09–2.71)	<0.001
	Very stressful	2.222	0.066	1133.210	9.23 (8.11–10.50)	<0.001
Subjective health status (ref = good)	Normal	0.192	0.069	7.798	1.21 (1.06–1.39)	0.005
	Bad	0.701	0.068	108.009	2.02 (1.77–2.30)	<0.001
Social support (ref = none)	≥1	−0.132	0.049	7.297	0.88 (0.80–0.97)	0.007
Changes in daily life due to COVID-19		−0.005	0.001	29.046	0.99 (0.99–0.99)	<0.001
Fear of COVID-19 infection		−0.014	0.023	0.359	0.99 (0.94–1.03)	0.549

## Data Availability

All data are available from the corresponding author upon request.

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
