# Peer review of "Predictors of Depression in Elderly According to Gender during COVID-19: Using the Data of 2020 Community Health Survey"

_healthcare, 2024, doi:10.3390/healthcare12050551_

Round 1

Reviewer 1 Report

Comments and Suggestions for Authors

The quality of the measures is rough and not nuanced limiting the conclusions that can be drawn from the results.  Very general measures without clarification into qualities such as type of social interaction, confidence in ability to access health care, or general anxiety over living conditions and having resources to weather problems that might occur leave open many possible explanations for the findings that could provide alternative meanings to the results.  A single item measure was used for most of the key variables that does not capture the complexity in any of them.  I noted the data came from health centers, do all residents make use of these centers or could there be biases in the population base of the data?

Comments on the Quality of English Language

Phrasing is formal and awkward at times.  Especially when presenting the findings the meaning of the numbers is not made clear and could cause some confusion.

Author Response

Thank you for your feedback.

1. I appreciate that you provided specific details about the limitations of the measurement tools.

Additionally, a limitation of this study is that it did not investigate beyond the frequency of social support to assess the context of interactions, such as whether they occurred face-to-face, non-face-to-face, or online.To assess the confidence in healthcare access, it can be considered to measure various aspects such as satisfaction with medical services, ease of access, and perception of the quality of healthcare services. In future research, it is necessary to include a variety of question types, such as Likert scales and open-ended questions, to deeply understand the experiences and perceptions of the participants. In this study, depression was assessed dichotomously; however, future research should consider using a variety of question formats to more thoroughly assess depression in the elderly. 

2.  The data were collected by utilizing personnel hired by the public health centers, who visited randomly selected households. The study did not target visitors to the health centers. Further explanations about the participants have been provided in the main text.

Therefore, the Community Health Survey is effective in collecting health-related information from various groups within the community, as it employs a random sampling method that includes diversity in age, gender, and socioeconomic status.

3. I made an effort to provide accurate and smooth translations

Thank you

Reviewer 2 Report

Comments and Suggestions for Authors

Congratulations on valuing the mental health of the elderly.

The topic itself is not new, and the causes of the decrease in signs of depression in older people are already widely known.

The added value of this study seems to me to be the scope of the population (total number of participants).

It does not seem appropriate for future conclusions to talk about new pandemics to define intervention programs for the prevention of depression in older people.

There could be some more objective recommendations in the conclusion for possible interventions for the future.

Author Response

Thank you for your feedback.

  1. I have removed the part about future prevention of new pandemics from the conclusion. Instead, I have provided more specific recommendations for future interventions

To support the elderly, it's essential to expand online platform-based counseling and education programs, and provide continuous support using telephone or messaging services. Additionally, developing programs that offer opportunities for active participation to prevent social isolation and strengthen social support networks among seniors is crucial. These programs should aim to enhance social networking and community engagement among the elderly.

   2. I made an effort to provide accurate and smooth translations

Thank you

Reviewer 3 Report

Comments and Suggestions for Authors

The study is well written. The introduction is fine, however, there is room to add hypotheses for the study.

The methods must be improved, in this case, I believe that the measures of depression and change in daily routines must be clarified. How did the authors test depression? what components of daily life have been changed? The existence of depression consisted of a yes/no answer question: ‘Have you ever 133 felt so sad or hopeless that it interferes with your daily life for more than two weeks in a  row over the past year?’  Is this valid to test depression?

I think that using network analyses me be more adequate for this kind of study. 

As the authors stated, "However, this study is a secondary data analysis  study based on existing community health survey data, and some limitations exist in the interpretation of its results." This limitation may make a bias and the results may be affected.

Is this the only limitation of the study?

I think that the study is good but I don't believe that it adds to the current knowledge. 

Comments on the Quality of English Language

Minor editing of English language required

Author Response

Thank you for your feedback.

  1. I have elaborated on the limitations of the research methods and suggested areas for improvement

Additionally, a limitation of this study is that it did not investigate beyond the frequency of social support to assess the context of interactions, such as whether they occurred face-to-face, non-face-to-face, or online.To assess the confidence in healthcare access, it can be considered to measure various aspects such as satisfaction with medical services, ease of access, and perception of the quality of healthcare services. In future research, it is necessary to include a variety of question types, such as Likert scales and open-ended questions, to deeply understand the experiences and perceptions of the participants. In this study, depression was assessed dichotomously; however, future research should consider using a variety of question formats to more thoroughly assess depression in the elderly. 

To support the elderly, it's essential to expand online platform-based counseling and education programs, and provide continuous support using telephone or messaging services. Additionally, developing programs that offer opportunities for active participation to prevent social isolation and strengthen social support networks among seniors is crucial. These programs should aim to enhance social networking and community engagement among the elderly.

   2. I made an effort to provide accurate and smooth translations

Thank you

Round 2

Reviewer 2 Report

Comments and Suggestions for Authors

Congratulations,

I think that the authors can relacionated the social isolament with the need of change of interventions.

Author Response

Dear Reviewer,

Thank you for your insightful comment suggesting that we explore the relationship between social isolation and the need for changes in intervention strategies. Your feedback prompted a thoughtful reconsideration of our approach to addressing the complexities of social isolation among the elderly, especially in the context of the COVID-19 pandemic.

In response to your valuable suggestion, we have revised our manuscript to emphasize the link between social isolation due to factors such as COVID-19 and its impact on depression among the elderly. We now argue that this connection underscores the necessity for a diverse range of policies aimed at mitigating social isolation and actively preventing depression within this demographic.

Furthermore, we acknowledged the insights from prior research indicating that a universal solution to loneliness and social isolation does not exist. Reflecting on this, our revised manuscript proposes that interventions should be customized to meet the unique needs of individuals, specific groups, or varying degrees of loneliness. Consequently, we have incorporated a discussion on the need for personalized interventions tailored to the specific needs and conditions of the elderly.

We believe these revisions address your concerns and enrich the manuscript by providing a nuanced understanding of the interplay between social isolation and intervention strategies. We appreciate the opportunity to enhance our work based on your feedback and hope that our revised submission meets your expectations.

Sincerely,

Kim K.M

Reviewer 3 Report

Comments and Suggestions for Authors

I thank the authors for the corrections they made.

Unfortunately, they did not answer in an organized manner the points I raised and it seems that the information is mixed, there was room to address point by point.

I'm still skeptical about the depression measure. I don't think it assesses depression. But this is at the discretion of the editor. In addition, I think it should be emphasized that this research adds to the existing knowledge but not anew.

Author Response

Thank you for your feedback.

Although this tool is not designed to assess the presence or severity of depression, which could be a limitation of this study, it is significant in that it checks whether elderly individuals have recently experienced feelings of depression and uses this as a basis to predict depression among the elderly during a pandemic. While this may not be a novel discovery, the significance of this study lies in moving beyond previous research that identified depression in the elderly, by distinguishing depression in the elderly according to gender. 

Thank you

Round 3

Reviewer 3 Report

Comments and Suggestions for Authors

Thank you..

I think that the manuscript can be published.

Author Response

Thank you